# Were domestic camelids present on the prehispanic South American agricultural frontier? An ancient DNA study

Cinthia Carolina Abbona[1]*, Gustavo Neme Adolfo[1], Jeff Johnson[2,4], Tracy Kim[2], Adolfo Fabian Gil[1], Steve Wolverton[3,4]

1 IDEVEA (CONICET-UTN, FRSR), San Rafael, Mendoza, Argentina, 2 Department of Biological Sciences, University of North Texas, Denton, Texas, United States of America, 3 Department of Geography and the Environment, University of North Texas, Denton, Texas, United States of America, 4 Advanced Environmental Research Institute, University of North Texas, Denton, Texas, United States of America

* abbonacinthia@gmail.com, cabbona@conicet-mendoza.gob.ar

**Data Availability Statement:** The raw data from NGS are available in the SRA accession: PRJNA603673, with the Submission ID: SUB6896752 Data Review URL is https://dataview.

## Abstract

The southern boundary of prehispanic farming in South America occurs in central Mendoza Province, Argentina at approximately 34 degrees south latitude. Archaeological evidence of farming includes the recovery of macrobotanical remains of cultigens and isotopic chemistry of human bone. Since the 1990s, archaeologists have also hypothesized that the llama (*Lama glama*), a domesticated South American camelid, was also herded near the southern boundary of prehispanic farming. The remains of a wild congeneric camelid, the guanaco (*Lama guanicoe*), however, are common in archaeological sites throughout Mendoza Province. It is difficult to distinguish bones of the domestic llama from wild guanaco in terms of osteological morphology, and therefore, claims that llama were in geographic areas where guanaco were also present based on osteometric analysis alone remain equivocal. A recent study, for example, claimed that twenty-five percent of the camelid remains from the high elevation Andes site of Laguna del Diamante S4 were identified based on osteometric evidence as domestic llama, but guanaco are also a likely candidate since the two species overlap in size. We test the hypothesis that domesticated camelids occurred in prehispanic, southern Mendoza through analysis of ancient DNA. We generated whole mitochondrial genome datasets from 41 samples from southern Mendoza late Holocene archaeological sites, located between 450 and 3400 meters above sea level (masl). All camelid samples from those sites were identified as guanaco; thus, we have no evidence to support the hypothesis that the domestic llama occurred in prehispanic southern Mendoza.

## Introduction

Southern Mendoza is thought to be the southern limit of the dispersion of prehispanic (>500 years ago) agriculture in South America [1–7]. Evidence of agriculture includes macrobotanical remains of domestic plants, the presence of human osteopathies that relate to an agricultural diet, and $\delta^{13}C$ and $\delta^{15}N$ signatures measured in human remains [4, 8, 9]. While the

ncbi.nlm.nih.gov/object/PRJNA603673?reviewer=r0stc5g54tm9ttaj0cp96n4klj.

**Funding:** This work was supported by United States National Science Foundation Grant number 1630051 (Principal Investigator – SW; Co-Principal Investigator – JJ; Co-Principal Investigator – LN) and Agencia Nacional de Promotion Cientifica PICT 2013-0881. The funders had no role in study design, data collection and analysis, decision to publish, or preparation of the manuscript. Additional funding was provided by the short-term mobility Fulbright scholarship and CONICET (CA).

**Competing interests:** The authors have declared that no competing interests

timing of the first arrival of cultigens to southern Mendoza is debated, it is estimated to be approximately 2,200 radiocarbon years before present. However, the initial presence of cultigens may not represent the onset of agriculture, and apparently during this early stage their contribution to the diet was not important [1–4, 10]. Traditionally, in South American archaeology, agriculture was associated with the presence of pastoral methods, and therefore domestic animals, such as llama (*Lama glama*) or alpaca (*Vicugna pacos*) [11–14]. Although the prehispanic presence of cultigens in southern Mendoza is supported by multiple lines of evidence, pastoral practices using domestic camelids have only recently been proposed based on archaeological remains from this area [13, 15–17]. The unequivocal presence of llama and pastoralism at this latitude is only established for the neighboring central Chile during the Inka empire between AD 1470 and 1536 [18–21].

Southern Mendoza province represents the northern limit of Patagonia from an environmental perspective [10, 22]. In this region, zooarchaeologists have identified wild camelid faunal remains as guanaco (*Lama guanicoe*) [23–31]. During the early Holocene, however, faunal remains of another wild camelid the vicuña (*Vicugna vicugna*), have also been identified at the Agua de la Cave site (Fig 1A) located in the Andes at 200 km north to Southern Mendoza [15, 24, 32].

Today, the guanaco is the only wild camelid in Patagonia, because the distribution of vicuña extends only as far south as northern San Juan Province, which is 500 km to the north (Fig 1B) [35, 36]. South American domestic camelids include llama and alpaca, but only llama is considered in this paper due to minimal evidence for the presence of alpaca from precolumbian sites in Argentina [37] and Chile [38, 39].

In addition to the morphometric identification of llama remains in the northern Mendoza archaeological record [15, 40, 41] its presence was assumed due to the late but strong presence of the Inka empire in the region [14, 33, 34]. In contrast, in southern San Juan province (500 km to the North), there are clear archaeological indicators of the presence of llama, such as

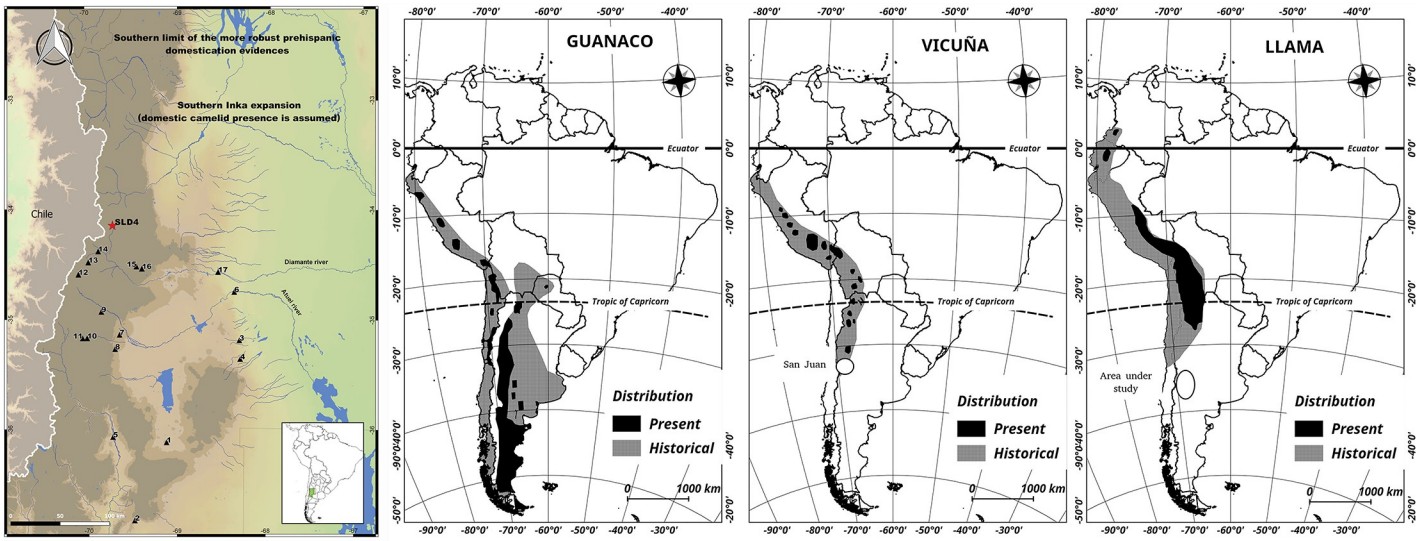

**Fig 1. Camelid distribution, boundaries and archaeological sites from the study area. A)** The map shows the boundary for accepted archaeological evidence of prehispanic domestication and Inka expansion near the study area [14, 33, 34]. Archaeological sites where ancient DNA samples were recovered: 1) Volcán El Hoyo; 2) Agua de Pérez; 3) Los Leones-6; 4) Agua de los Caballos; 5) Cueva de la Luna; 6) Zanjón El Morado; 7) Ojo de Agua; 8) Cueva Salamanca; 9) Cueva Palulo; 10) Arroyo El Desecho-4; 11) Cueva Arroyo Colorado; 12) Los Peuquenes; 13) El Indígeno; 14) Risco de los Indios; 15) El Carrizalito; 16) Alero Montiel; 17) Fuerte San Rafael del Diamante. LD-S4 site (red star) where llama was identified using morphometric analyses. High elevation sites near to LD-S4 site: Los Peuquenes (12), El Indigeno (13), and Risco de los Indios (14). The map uses NASA open data (https://earthdata.nasa.gov). **B)** The historic and present distribution of guanaco, llama and vicuña from Franklin [35].

dung deposits, textiles, and a large number of depictions in rock art, pottery, and "furniture art" [42–45]. The southernmost evidence of Inka occupation on the eastern slope of the Andes has been proposed for the Agua Amarga site in Tunuyán, located 100 km to the north of the study area [46]. The distribution of the Inka empire was similar on the western side of the Andes (central Chile), with an absence of pre-Inka evidence of domestic camelids at this latitude in the archaeological record [47–49].

Despite the absence of archaeological and ethnohistorical evidence, some researchers argue that prehispanic farming groups in the northern limit of Patagonia were camelid herders [13, 16, 17, 50]. These arguments focus on the presence of remains of domestic plants and pottery at archaeological sites with camelid herding inferred by extrapolation. It is important to note, however, that plant domestication, pastoral practices, and the adoption and development of pottery technology do not necessarily occur contemporaneously in many areas of the world [51–54]. Other inferences of camelid herding in southern Mendoza is based on morphometric analysis of camelid osteological remains [15], leading to the claim that domesticated llama, specifically, were present in the region. This claim is based on analysis of zooarchaeological remains from Laguna del Diamante-S4 (LD-S4) site (Fig 1A) from archaeological contexts dated from 700 to 950 years BP [15]. The identification of the remains as llama as opposed to guanaco was based on osteometric analysis indicating large body size, as llamas were intentionally bred for increased body mass. At LD-S4, 25% of the total faunal assemblage (n = 12 samples) was identified as llama. Gasco [15] identified a similar proportion of llama from zooarchaeological assemblages from northern Mendoza and central and southern San Juan using the same morphometric criteria.

Claims that prehispanic llama and "herders" occurred south of 34° latitude have important implications for archaeological interpretations of past human lifeways in the region. The presence of such groups would imply changes in subsistence and settlement systems including reductions in residential mobility, technological and dietary changes, as well as a differentiated materiality from the previous occupations [40, 55–57]. If the presence of herder groups in the northern limit of Patagonia is confirmed, it would also require revision of many interpretations of how people used the landscape, interpretations of human demography, and conclusions about human impacts on the environment, all of which would be dramatically different if herding societies were present in those areas. The morphometric-based identification of camelid bones as llama [15] is the only empirical evidence used to support such claims.

The use of morphometry to study domestication has been widely used in South America [39, 40, 57–62]. Even though morphometry has many advantages, researchers recognize the limitations of the approach, especially for making inferences concerning past geographic range extension based on small sample sizes [38, 59–62]. In contrast, analysis of ancient mitochondrial DNA (aDNA) generated from zooarchaeological samples can provide a stronger approach for species identification including population-level comparisons in current and archaeological contexts when species identification is less conclusive based on morphology [63–67]. Analysis of DNA from domestic and wild species can also help delineate domestication events [68–70]. Using DNA to study South American camelids, Marin et al. [69, 71] determined that the llama was domesticated from guanaco and the alpaca from vicuña. Analysis of aDNA can also be used to assess whether llamas were present in southern Mendoza/northern Patagonia, which would indicate the presence of a pastoral society in the region and subsequently extend the southernmost limit of prehispanic agriculture in the Americas.

In this paper, we employ analysis of aDNA as a method for identifying species among camelid remains from archaeological sites in Mendoza. We analyze whole mitochondrial DNA genomic sequence data generated from 50 archaeological samples of camelid bone from central and southern Mendoza, 41 of which produced sufficient aDNA for analysis. Remains from

LD-S4 were not analyzed in this study because we were not able to obtain permission. Diagnostic nucleotide positions within the mitochondrial genome were used to assign samples to species based on taxon-specific DNA sequence data publicly available on GenBank (www.ncbi.nlm.nih.gov/genbank). This approach made it possible to assess whether any of the archaeological samples support the hypothesis that domestic llamas were present in prehispanic southern Mendoza.

## Materials and methods

Fifty camelid bone samples (identified as *Lama* spp. based on osteological characters) from 25 cultural assemblages excavated from 17 archaeological sites located in central and southern Mendoza (Fig 1A) were selected for analysis. This represents 70% of the available archaeological sites with faunal remains from the region. Each sample was assigned to a temporal assemblage using associated radiocarbon dates and artifact types [28–30, 72]. Several types of sites are represented including base camps, kill sites, butchery stations, and small camps occupied for short periods. Contexts include open-air sites, rockshelters, and caves (see Table 1). Some of the sites have remains of domestic plants and/or association with human remains with values of $\delta^{13}C$ that suggest corn consumption [73].

DNA extraction and library preparation were conducted in a dedicated aDNA laboratory at the University of North Texas using methods developed specifically to minimize contamination with contemporary DNA. It is also important to note that no camelid samples had been processed in the lab prior to this study. Genomic DNA extraction followed methods originally described elsewhere [74] with minor modifications as also described in [75, 76]. All tools and bench-top surfaces were cleaned with 10% bleach and 95% ethanol between each sample prep, and the surface of each bone sample was sterilized prior to tissue extraction using a UV cross-linker CL-1000 (UVP, Upland, CA, USA) for 10 minutes (5 minutes per side). All bone samples were also cleaned by removing approx 1mm of surface layer using a sterile drill and disc in an airflow hood in a separate room from aDNA extraction. Dense interior cortical bone was removed and pulvurized to a fine powder using a SPEX 6775 Freezer/Mill cryogenic grinder (SPEX SamplePrep, Metuchen, NJ, USA).

The aDNA laboratory is dedicated solely for work on DNA extraction from ancient samples. Extractions were conducted in groups of fifteen that included a negative control in each group to verify no cross-contamination or reagent contamination. Approximately 150 mg of bone powder from each sample was pre-digested for 1 h at 56°C in 1 mL of lysis buffer (Proteinase K 0.25 mg/ml, N-laurylsarcosyl 0.5%, EDTA 0.45 M, pH8) to limit contemporary bacterial contamination. After pre-digestion, samples were centrifuged at 500 xg for 5 minutes and the supernatant was removed. The undigested bone pellets were again incubated in 1mL of extraction buffer overnight at 37°C. After the second digestion, samples were centrifuged for 5 min at 500 xg and the supernatant was mixed with 3 ml Tris-EDTA (TE) buffer 1x, in an Amicon Ultra 4 30 kDa centrifugal filter (EMD Millipore) and centrifuged for 10 min at 4,000 xg. The filter was washed with 2 ml TE and centrifuged for 8 min at 4,000 xg, the remaining 50 ul of the sample in the filter was brought to a total volume of 100ul. The final purification step was carried out with a QIAquick column (Qiagen) and eluted in 45ul with elution buffer (EB, Qiagen).

Total DNA was treated with 1X USER enzyme mix (New England BioLabs) for 3 h at 37°C to reduce nucleotide substitution errors associated with cytosine deamination, which is common when working with aDNA [77–79]. DNA concentrations of all extracted samples were measured using a Qubit Fluorometer with the dsDNA HS Assay Kit (Invitrogen, Carlsbad, CA, USA). Sequencing libraries were then generated with a starting concentration of

**Table 1. Assemblage and site descriptions for the samples.**

| Site | Sample ID | Funcionality | Kind of site | Years BP | skeletal element |
|---|---|---|---|---|---|
| Agua de los Caballos | 14 | undetm | Cave | 300 | Metapodial |
| Agua de Perez | 87 | BC | Open air | 685 | Mandible |
| Agua de Perez | 88* | BC | Open air | 685 | Ischium |
| Alero Montiel | 17* | undetm | Rockshelter | 1800 | Phalanx |
| Alero Montiel | 39 | undetm | Rockshelter | 2240 | Tooth |
| Cave Arroyo Colorado | 5* | SAS | Cave | 770 | Astragalus |
| Cave Arroyo Colorado | 6 | SAS | Cave | 770 | Carpal |
| Cave Arroyo Colorado | 33 | undetm | Cave | 770 | Metapodial |
| Cave de Luna | 20 | undetm | Cave | 1400 | Phalanx |
| Cave de Luna | 38 | undetm | Cave | 500 | Tooth |
| Cave Palulo | 19 | SAS | Cave | 2050 | Humerus |
| Cave Palulo | 61 | SAS | Cave | 130 | Scapula |
| Cave Palulo | 63 | SAS | Cave | 130 | Carpal |
| Cave Palulo | 65 | SAS | Cave | 2030 | Long bone shaft |
| Cave Salamanca | 31* | undetm | Cave | 1500 | Metacarpal |
| Cave Salamanca | 72 | undetm | Cave | 2200 | Phalanx |
| Cave Salamanca | 85* | undetm | Cave | 2200 | Metapodial |
| Cave Salamanca | 86 | undetm | Cave | 2200 | Magnum |
| Cave Salamanca | 94 | undetm | Cave | 2200 | Metapodial |
| Cave Salamanca | 95 | undetm | Cave | 2200 | Phalanx |
| Cave Salamanca | 96 | undetm | Cave | 2200 | Tibia |
| Cave Salamanca | 97 | undetm | Cave | 1360 | Metapodial |
| Cave Salamanca | 102 | undetm | Cave | 7000 | Phalanx |
| El Desecho 4 | 45 | BC | Open air | 5500 | Phalanx |
| El Desecho 4 | 47* | BC | Open air | 5500 | Metapodial |
| El Desecho 4 | 49* | BC | Open air | 5500 | Long bone fragment |
| El Indigeno | 106 | BC | Structures | 900 | Phalanx |
| El Indigeno | 107 | BC | Structures | 900 | Metapodial |
| El Indigeno | 108 | BC | Structures | 900 | Metapodial |
| El Perdido 4 | 121 | BC | Open air | 2600 | Third Phalanx |
| El Perdido 5 | 119 | BC | Open air | 2100 | Cuneiform |
| Fuerte SRD | 66 | undetm | Historico | 200 | Metapodial epiphysis |
| Fuerte SRD | 67 | undetm | Historico | 200 | Proximal metacarpal |
| Fuerte SRD | 71 | undetm | Historico | 200 | Ulna |
| Fuerte SRD | 110 | undetm | Historico | 200 | Phalanx |
| Fuerte SRD | 112 | undetm | Historico | 200 | Metapodial |
| Fuerte SRD | 113 | undetm | Historico | 200 | Metapodial |
| Gruta Carrizalito | 92 | undetm | Cave | 530 | Cuboid |
| Los Leones 6 | 78 | undetm | Open air | 300 | Calcaneus |
| Los Leones 6 | 79 | undetm | Open air | 300 | Metatarsal distal |
| Los Peuquenes | 25 | BC | Structures | 360 | Phalanx |
| Los Peuquenes | 80 | BC | Structures | 360 | Carpal |
| Ojo de Agua | 109 | BC | Open air | 200 | Metapodial |
| Puesto Ortubia | 59* | BC | Open air | 900 | Long bone shaft |
| Risco de los Indios | 53 | BC | Structures | 500 | Phalanx |
| Volcan El Hollo | 91* | SAS | Cave | 500 | Second Phalanx |
| Volcan El Hollo | 100 | SAS | Cave | 500 | Metapodial |

(*Continued*)

**Table 1.** (Continued)

| Site | Sample ID | Funcionality | Kind of site | Years BP | skeletal element |
|---|---|---|---|---|---|
| Zanjon Morado | 81 | SAS | Rockshelter | 1200 | Phalanx (juv) |
| Zanjon Morado | 82 | SAS | Rockshelter | 1200 | Phalanx |
| Zanjon Morado | 83 | SAS | Rockshelter | 1200 | Phalanx distal |

Notes: Of the 50 samples, adequate sequence data coverage was obtained for further analysis for 41 samples. BC (base camp), SAS (specific activities site), undetm (undetermined). The date of each bone sample was assigned by direct association with radiocarbon dates from charcoal. The exceptions are the samples from Fuerte SRD (date is based on historical documents) and Ojo de Agua (date is based on cultural material association). The samples are available in the Museo de Historia Natural de San Rafael, Archaeology lab. No permits were required for the described study, which complied with all relevant regulations

(*) These samples were not used for analysis due to low coverage.

approximately 1.95 ug/ml following a modified version of the Blunt-End Single-Tube method (BEST; [80]) to allow for double-indexing. Amplified libraries were validated using the Qubit Fluorometer and the Agilent 2100 Bioanalyzer (Agilent Biosystems, Santa Clara, CA, USA). The libraries were also enriched for mitochondrial fragments following the procedure described by Maricic et al. [81], with minor modifications using a predesigned *Lama glama* myBaits Mito panel (ArborBioSciences, Ann Arbor, MI, USA). All of the libraries were pooled and sequenced using a 150 cycles reagent cartridge (2x75) on an Illumina NextSeq 500 sequencing platform (Illumina, San Diego, CA, USA) at University of North Texas BioDiscovery Institute Genomics Center (Denton, TX, USA), targeting an initial 200,000 paired end reads (clusters) per library. The raw data are available in the National Center for Biotechnology Information Sequence Read Archive: SRA accession PRJNA603673.

After sequencing, paired-end reads were filtered based on quality and mapped to the guanaco mitochondrial genome (GenBank accession: NC011822). Prior to mapping, SeqPrep (https://github.com/jstjohn/SeqPrep) was used with default settings to trim adapters and merge overlapping paired-end reads. Merged reads were then aligned to the reference mitochondrial genome using Mitochondrial Iterative Assembler (MIA) [82], using a kmer filter of length 13 (-k 13). After mapping, the consensus mitochondrial (mtDNA) genome sequences were determined with a minimum of 3x coverage per base and 2/3 of those bases in agreement. Sites not meeting those criteria were identified as missing. The final mtDNA alignment (16,649 base-pairs (bp)) was created using default parameters of ClustalW Alignment [83], as implemented in Geneious v.7.1.9 [84]. The mtDNA sequence dataset was then reduced to two loci, specifically cytochrome *b* (cytB 1,140 bp) and the D-loop (1,215 bp), to allow the inclusion of additional samples available on GenBank to perform species identification of the aDNA samples. The two loci were chosen not only because the majority of mtDNA sequences available on GenBank for the three focal taxa were either cytB or D-loop, but also because the number of nucleotide substitutions for the two loci differ by at least 5% and 8%, respectively, between guanaco and domestic llama or vicuña based on all available mtDNA sequences for those two loci at the time of this study. The latter point is important when working with aDNA because their final consensus DNA sequences may include ambiguous bases (i.e., unknown nucleotide sites, or Ns) after sufficient quality filtering has been achieved depending on a nucleotide sites' overall depth of coverage. Having reduncancy in the number of similar or differing bases between samples along a DNA alignment is important to ascertain species identification of unknown samples.

Phylogenetic relationships were reconstructed with mtDNA cytB and D-loop sequence data for a total of 71 individuals, including 41 ancient sequences analysed in this work and 30 previously sequenced ancient (n = 3) and contemporary (n = 27) samples of guanaco, llama

and vicuña available on GenBank (S1 Table). Phylogenetic reconstruction was performed using a Bayesian method implemented in BEAST [85] and Maximum Likelihood (ML) using Garli 0.951 [86].

BEAUti was used to prepare alignments for phylogenetic tree reconstruction using BEAST. The model of DNA substitution that best fit the data was identified using a hierarchical likelihood ratio test and Akaike information criterion as implemented in the program MODELTEST 3.7 [87]. The model that best fit the cytB data was the HKY+G model, and the HKY+I+G model was identified for the D-loop sequences and a Speciation: Birth-Death process tree prior [88]. The Markov Chain Monte Carlo (MCMC) was run from a random starting tree for 20,000,000 iterations, sampling every 1,000th tree with a burn in of 200,000 states. The effective sample size for estimated parameters exceeded 200, which was verified using Tracer [89]. Posterior probabilities were annotated onto the BEAST output tree using TreeAnnotator. Maximum likelihood analyses were performed with Garli 0.951 [86] under the substitution model HKY+G+I. One hundred bootstrap (BS) replicates were performed.

## Results

A total of 50 samples were sequenced, of which aDNA from the mitochondrial genome was recovered from 41 samples from 17 archaeological sites (Fig 1A). Illumina sequencing reads from each sequencing library were mapped to the guanaco, llama and vicuña mitochondrial reference genomes to assess potential ascertainment bias. In each case a higher depth of coverage was achieved when mapped with the guanaco mtDNA reference genome and used for subsequent analyses (average depth of coverage = 461, range = 9.4 to 2098.4, S2 Table). Moreover, the consensus sequences from different reference genomes were identical independent of whether the sequencing reads were mapped against guanaco, llama or vicuña. Phylogenetic analyses were conducted with the consensus sequences mapped to the guanaco due to the higher coverage mappings. The Maximum Likelihood (ML) and Bayesian phylogenetic reconstructions yielded a well-resolved tree topology, with all 41 archaeological samples placed unambiguously within the guanaco clade with high bootstrap support (>91) and posterior probabilities (>0.99) (Fig 2).

The ancient Holocene samples were identified as *Lama guanicoe* based on their placement within the resulting concensus trees. As shown previously using mtDNA data [90], guanaco form a monophyletic clade separate from vicuña and domestic llama based on mtDNA cytB and D-loop loci, making species identification of the bone samples possible.

## Discussion

This study of ancient camelid mtDNA identified the remains of prehispanic camelids from archaeological sites in central and southern Mendoza as guanaco. Multiple phylogenetic analyses using cytB and D-loop grouped the southern Mendoza camelid samples with contemporary guanaco, to the exclusion of domestic llama. These results are consistent with previous analyses based on morphological data, which also identified the southern Mendoza camelids from the archaeological sites as guanaco [10, 23–26, 28–31].

The results of this study do not confirm the presence of domestic camelids in southern Mendoza. Such is the case in a broad sense in that the 41 samples geographically represent ecosystems in the region, particularly for samples dating to the last 2,000 years BP when domestic plants arrived in southern Mendoza. Archaeologists have argued that prehispanic pastoralist societies occupied southern Mendoza by 2,000 years ago [13, 16, 17]. This argument is based on evidence of domestic plant remains and pottery technology from archaeological sites in the southern boundary of agricultural societies. However, such indicators do not necessarily

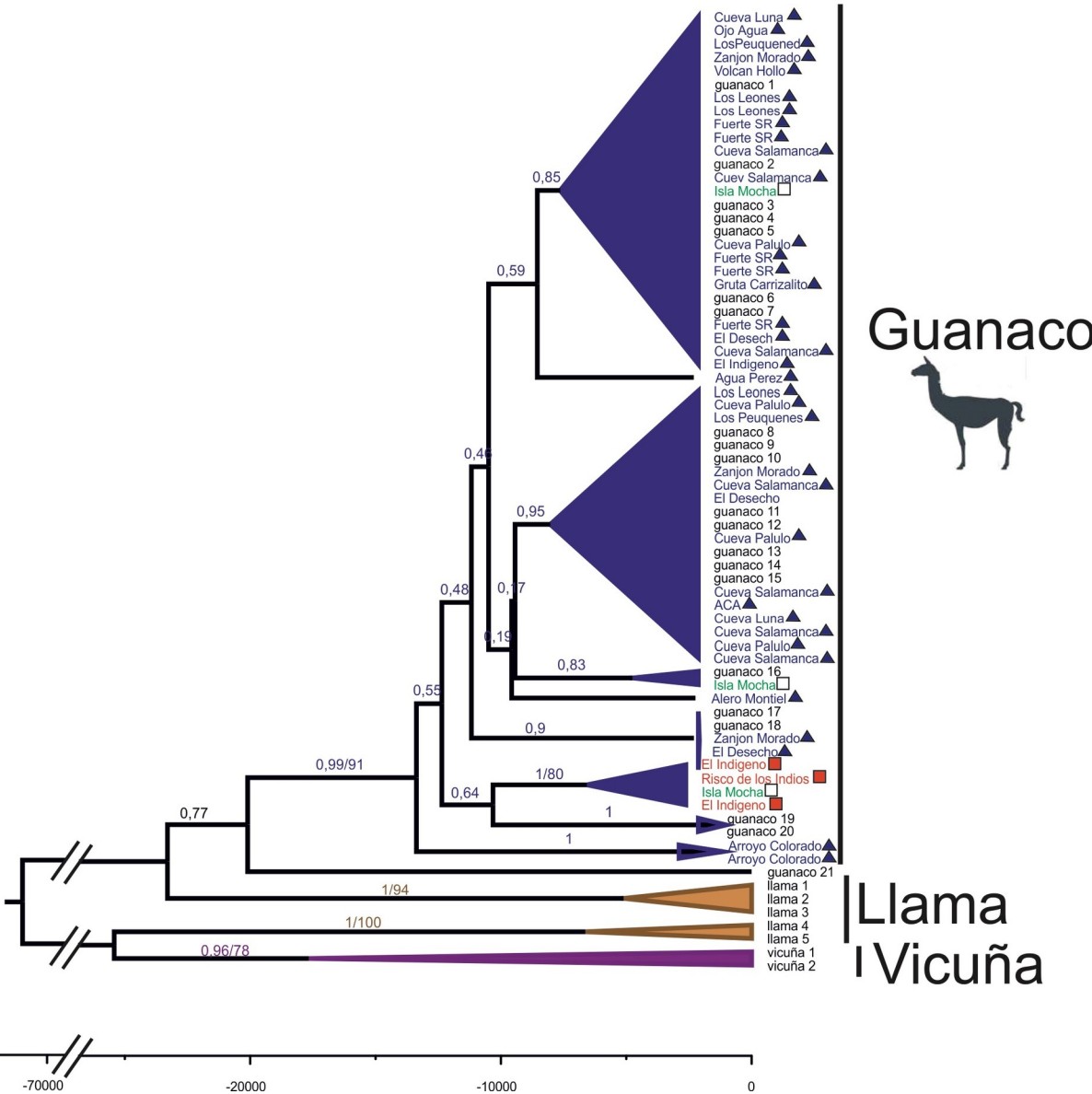

**Fig 2. Phylogenetic relationship of South American camelids.** The 50% majority rule consensus tree results from the Bayesian analyses of the cytB and D-loop sequence dataset. Nodal support values represent the Bayesian posterior probabilities/maximum likelihood bootstrap values (1,000 replications). Clades with nodal support values below that 0.5 or 50% have been collapsed. As currently presented, the guanaco clade is one large collapsed polytomy. Sample names are color coded according to whether the ancient sample was sequenced for this study (blue and filled triangle) and originated near the high-altitude LD-S4 site (red and filled square), or were ancient (green and empty square) or contemporary (black and without symbol) mtDNA sequences obtained from GenBank.

provide support for prehispanic pastoralism. On the contrary, such evidence may be present without animal domestication as has been shown in different parts of the world [51–54].

Southern Mendoza has been characterized as the agricultural dispersion boundary in South America with human societies to the north developing agricultural methods for subsistence starting two thousand years ago and those to the south practicing hunting and gathering prior to Spanish colonization approximately 500 years ago [1–4, 6, 7, 73, 91]. This boundary extends from west to east along the Atuel and Diamante rivers at 34° 40' south latitude near the Patagonia phytogeographic boundary. The specific location of the transitional zone between these

two populations is unclear and has fluctuated over time depending on archaeological evidence [4, 5, 25, 32, 73]. The results from this study do not indicate that pastoralism was present south of the agricultural dispersion boundary.

The absence of herding, however, does not mean that people did not practice agriculture south of the dispersion boundary. The southernmost archaeological evidence of prehispanic farming in Argentina is the presence of remains of *Zea mays*, *Phaseolus vulgaris*, *Cucurbita pepo*, *Lagenaria siceraria* and *Chenopodium quinoa* [2, 3]. Corn (*Zea mays)* has been considered the most important crop for farming because of its ubiquity in the archaeological record. Dependence on crops, however, was minimal and variable with limited consumption that was restricted to specific biogeographic contexts, such as sites located in Monte desert area [73]. There is no evidence of irrigation or semi-permanent or perminent housing structures suggesting that crops served a minor role in prehispanic economies in the Southern Mendoza region [4, 32].

The archaeological site of Laguna del Diamante-4 (LD-S4)—where researchers claimed to have identified specimens of domesticated llama using morphometric analysis—is one of several high elevation camps located from 2,400 to 3,400 meters above sea level (masl) [92, 93]. LD-S4 is located near the sources of the Atuel and Diamante rivers, in Laguna El Diamante, near the boundary between Argentina and Chile. The most important features of these high elevations sites include: the presence of semicircular stone housing structures ("pircados"), high frequencies of fragmented pottery, strong evidence for the consumption of guanaco, with chronologies dating back to the last 2,000 years BP in Argentina and ca 3,000 years BP on the western slope of the Andes [26, 92–94]. Until recently, these sites were thought to represent hunter-gatherer occupations, but Gasco [15] argued that they could also have been occupied by herders based on the purported identification of camelid remains as llama. As with other biogeographic contexts south of the agricultural dispersion boundary, our results do not support that llama herding occurred in these high elevation contexts.

The biogeography and population biology of contemporary camelids in southern Mendoza also does not support the presence of llama in the region. Today, South American camelids include the wild species guanaco and vicuña and the domestic species llama and alpaca. In Argentina, contemporary guanaco populations are spread throughout Patagonia as wild and semicaptive ranching populations. The four *Lama* and *Vicugna* camelid species each have 72 chromosomes [95] and possess very similar C and G banding patterns [96] so it is difficult their differentiation from cytogenetically. Sequencing methods are required for distinguishing members of *Lama*, which are difficult to distinguish based on morphology of skeletal remains; the same is also the case for separating *Lama* spp. and *Vicugna* spp. [69], which highlights the importance of our results for investigating the late Holocene biogeographic distribution of camelids near the southern agricultural dispersion boundary.

Indeed, the northern guanaco population, *L. g. cacsilensis*, represents the parental population of the lineage that led to llama [71]. Analysis of a male-specific Y-chromosome marker in the genus *Lama* supports that there were independent domestication events of llama from guanaco and of alpaca from vicuña. This evidence is based on the major DBY patriline haplotypes, which originated prior to domestication. The maternal lineage divergence among vicuña–alpaca is greater than between guanaco and llama based on mitochondrial DNA [71]. However, it has proven difficult to describe the phylogenetic relationship among wild and domestic camelids due to extensive hybridization between llamas and alpacas and their near extirpation during the Spanish conquest [69, 97, 98]. Thus, the results of this study help clarify the Holocene population biology history of camelids in the region.

While hybridization among all four species can occur, previous research suggests that it is less likely to occur in the wild because shared haplotypes are uncommon between guanacos

and vicuñas [71], which may indicate the presence of a reproductive barrier [99]. Analysis of the partial or complete sequence of the cytB gene and the control region [69, 98, 100], as well as analysis of the mitochondrial genome [90] have successfully resolved the identification of South American camelids. Phylogenetic analysis of the aDNA sequences presented here grouped all the samples in the same clade as guanaco. The phylogeny also showed a monophyletic group between high elevation individuals and an archaeological individual from Mocha Island in Chile. The results of our study indicate not only that remains from archaeological sites represent guanaco, but that there is no evidence of interbreeding among various camelid species in the region.

A limitation of this study is that sampling may not be extensive enough to represent camelid diversity in the region. However, our study includes samples from a comprehensive variety of environments and periods, from both prehispanic and Hispanic sites (Table 1) that include periods when agriculture was present in the region, yet none of the camelid samples from these contexts is identifiable to domestic llama based on their aDNA.

Previous osteometric analyses of bone specimens from a single high elevation village from southern Mendoza (Site LD-S4; [15]) appear to support the presence of domestic camelids in this region. While we were unable to include these samples in our study, six DNA samples from camelids obtained from three other high elevation sites (El Indígeno, Los Peuquenes and Risco de los Indios) that are in close geographic proximity to LD-S4 and that date to approximately the same period [26, 72] were included in the analysis. The archaeological record of these sites suggests a similar livelihood among their occupants, particularly in terms of subsistence strategies indicating that occupants probably belonged to the same socio-environmental system as the occupants of LD-S4 [26, 72, 92, 93]. All high elevation camelid bone samples strongly grouped with the guanaco samples instead of domestic llama (Fig 2). In addition, camelid remains from the high elevation sites of Risco de los Indios and El Indígeno exhibited a very high similarity with an archaeological sample from Isla Mocha in Central Chile (Fig 2). There is high statistical support for the similarity despite the geographic distance between sites. As has been found in this study, camelid remains from Isla Mocha have also been identified as guanaco [90].

Our results weaken the claim that domestic camelids were present in prehispanic, high elevation contexts, further suggesting that size may not be a reliable indicator alone for distinguishing guanaco and domestic llama skeletal remains. The conclusion that the identified remains were from guanaco hunted in the proximity of the site is similar to results of other aDNA and zooarchaeological analyses focused on camelid remains located at different archaeological sites at similar latitude and elevation in central Chile and Argentina [4, 47, 48, 90, 93, 94, 101].

Whether pastoralists were present during prehispanic periods in the region as others have proposed has important implications for southern Mendoza archaeology. Such claims should be assessed using multiple lines of evidence. We argue that the presence of agricultural activities is not sufficient evidence to support that pastoralism was adopted. Morphometric identification of domestic camelids should be questioned, particularly related to studies of the archaeological record in boundary areas, such as central Chile and Argentina. In addition to the absence of domestic camelid DNA in our study, another weakness of the claim that domestic llama was present at LD-S4 is that the area would only have been habitable by people and camelids during a few summer months. Such environmental conditions would not have supported year-round pastoralism because of high snowfall during the winter. As a result, there should be evidence of pastoralist sites located below 2,000 meters, where pasture would have been available during the winter. However, no corral structures, dung deposits, or semi-permanent camps have been found. On the contrary, only hunter-gatherers activities have been documented [26, 44, 94, 102–104].

## Conclusion

In this study, we analyzed aDNA from camelid faunal remains from numerous sites from throughout southern Mendoza that date to the late Holocene. Bone samples from 41 individuals were identified as guanaco, and none of the remains were identified as llama based on phylogenetic analyses of aDNA including a subset of samples located near the LD-4 site that date to the same period as those reported in Gasco [15]. The results of our analysis support previous aDNA results from the western Andes in Chile, which also did not detect the presence of llama among camelid samples that were subsequently identified as guanaco [90]. The results of our analysis also support previous archaeological conclusions that there was no herding livelihood during the prehispanic period near the southern limit of farming. There is no evidence of corrals, artifacts that indicate herding technology, dung deposits, and other changes in material culture that would support the hypothesis that pastoralism was adopted. The results of our study and of previous ones indicate that despite that people adopted farming in some areas, insufficient evidence exists to support the presence of a pastoral lifestyle during the prehispanic period in southern Mendoza.

## Supporting information

**S1 Table. mtDNA sequences obtained from GenBank.**
(XLSX)

**S2 Table. Sequence data.** Percentages of total reads, endogenous DNA, identity and coverage of 41 consensus sequences.
(XLSX)

## Acknowledgments

We thank Amy Eddins, Garrett Meeks, and Summer Sherrod for their assistance with lab work and Dario Soria for his help with Fig 1B. We acknowledge the use of imagery provided by services from NASA's Global Imagery Browse Services (GIBS), part of NASA's Earth Observing System Data and Information System (EOSDIS). We thank reviewers who made a critical reading and have provided important observations which improved the manuscript.

## Author Contributions

**Conceptualization:** Cinthia Carolina Abbona, Gustavo Neme Adolfo, Jeff Johnson, Steve Wolverton.

**Data curation:** Cinthia Carolina Abbona, Jeff Johnson.

**Formal analysis:** Cinthia Carolina Abbona, Gustavo Neme Adolfo, Jeff Johnson, Steve Wolverton.

**Funding acquisition:** Gustavo Neme Adolfo, Jeff Johnson, Steve Wolverton.

**Investigation:** Cinthia Carolina Abbona, Gustavo Neme Adolfo, Jeff Johnson, Steve Wolverton.

**Methodology:** Cinthia Carolina Abbona, Jeff Johnson, Tracy Kim.

**Project administration:** Gustavo Neme Adolfo, Steve Wolverton.

**Resources:** Steve Wolverton.

**Supervision:** Gustavo Neme Adolfo, Jeff Johnson, Steve Wolverton.

**Validation:** Jeff Johnson, Steve Wolverton.

**Writing – original draft:** Cinthia Carolina Abbona, Gustavo Neme Adolfo, Jeff Johnson, Adolfo Fabian Gil, Steve Wolverton.

**Writing – review & editing:** Cinthia Carolina Abbona, Gustavo Neme Adolfo, Adolfo Fabian Gil, Steve Wolverton.

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
