## [Decision Letter · Decision Letter 0]

19 May 2020

PONE-D-20-06090

Were domestic camelids present on the preHispanic South American agricultural frontier? An ancient DNA Study

PLOS ONE

Dear Abbona,

Thank you for submitting your manuscript to PLOS ONE. After careful consideration, we feel that it has merit but does not fully meet PLOS ONE’s publication criteria as it currently stands. Therefore, we invite you to submit a revised version of the manuscript that addresses the points raised during the review process.

We would appreciate receiving your revised manuscript by Jul 03 2020 11:59PM. To enhance the reproducibility of your results, we recommend that if applicable you deposit your laboratory protocols in protocols.io, where a protocol can be assigned its own identifier (DOI) such that it can be cited independently in the future. For instructions see: http://journals.plos.org/plosone/s/submission-guidelines#loc-laboratory-protocols

We look forward to receiving your revised manuscript.

Kind regards,

David Caramelli, Ph.D

Academic Editor

PLOS ONE

Journal Requirements:

2. In your manuscript, please provide additional information regarding the specimens used in your study. Ensure that you have reported specimen numbers and complete repository information, including museum name and geographic location.

For more information on PLOS ONE's requirements for paleontology and archaeology research, see https://journals.plos.org/plosone/s/submission-guidelines#loc-paleontology-and-archaeology-research.

'We thank the Agencia Nacional de Promotion Cientifica PICT 2013-0881 and short-term mobility Fulbright scholarship.'

'This work was supported by United States National Science Foundation Grant number 1630051 (Principal Investigator - S. Wolverton; Co-Principal Investigator J. A. Johnson; Postdoctoral Researcher C.C. Abbona).'

5. We note that Figures 1 and 2 in your submission contain map/satellite images which may be copyrighted.

a. You may seek permission from the original copyright holder of Figures 1 and 2 to publish the content specifically under the CC BY 4.0 license. 

Reviewers' comments:

Reviewer's Responses to Questions

**Comments to the Author**

1. Is the manuscript technically sound, and do the data support the conclusions?

Reviewer #1: Yes

Reviewer #2: Yes

2. Has the statistical analysis been performed appropriately and rigorously? 

Reviewer #1: Yes

Reviewer #2: Yes

3. Have the authors made all data underlying the findings in their manuscript fully available?

Reviewer #1: Yes

Reviewer #2: Yes

4. Is the manuscript presented in an intelligible fashion and written in standard English?

Reviewer #1: Yes

Reviewer #2: No

5. Review Comments to the Author

Reviewer #1: The manuscript presents an interesting contribution to understand the domestication patterns in South America. It is well written, I have to signal only a couple of typos: line 90 "presence" and S1 Table "Provenience" (I suppose).

The study is based on a very good sampling for number of specimens, site representation and type of sites.

The authors choose to prepare UDG-treated libraries to be more confident with SNP calling. Even if I would be interest in knowing more details about the degree of preservation of the DNA in correlation with the age of the samples and altitude of the sites (and it is not possible to see the misincorporation patterns with this kind of library), I recognise that for the context of the study it is not strictly necessary to this detail to authenticate the results. Anyway, I would like to see in S1 Table at least the median length of the sequenced molecules mapping on the target.

I also suggest to spend some words to briefly describe the DNA extraction method, since it is not clear only through the cited reference.

It is a pity that the authors could not analyse any sample from Site LD-S4 that they mentioned as the one with evidence of domestic camelids according to osteometric evidence. Were those samples not available for the analysis?

Reviewer #2: The manuscript entitled “Were domestic camelids present on the preHispanic South American agricultural frontier? An ancient DNA Study” describes the application of ancient DNA analyses to clarify the past presence of domestic camelid (Lama glama) at the southern boundary of prehispanic farming in South America. The topic and the results are very interesting in the perspective of spread of llama domestication in this peculiar area but also in the perspective to add NGS data of ancient samples to databases, that will help to better analyse llama domestication in the past with a deep resolution respect to modern data.

The study is well developed, using cutting-edge NGS technologies and approriated phylogenetic analyses. However, there are some issues that need to be addressed:

1) In the abstract it would be better to indicate the number of samples for which the authors obtained results (41?) and not the total samples analysed (52?), or at least to specify that.

2) In Table 1 I think that the authors should: (i) add a colum where specify what kind of skeletal element was analysed (e.g. petrous bone, long bone, tooth...); (ii) add a column where they indicate if the chronology is based on radiocarbon dating or material culture; (iii) check the number of samples, because in the abstract and results the total number of samples is 52 and the ones with sufficient endogenous is 41, but in the table 1 there are 53 samples, of which 10 with sufficient endogenous; (iv) explain why some samples have the same ID, are they referred to the same individual?

3) The authors followed proper ancient DNA guidelines, but I suggest to add the results from MapDamage software to analyse damage patterns with the aim to authenticate the data here obtained.

4) Given that the identification of domestic from wild species through DNA analysis is often complicated and confused because of recent times of divergence and/or hybridisation phenomena, the paper will benefit from some more sentences about the power of mitochondrial DNA analysis in the identification of Lama glama and Lama guanicoe. For example, the authors stated that "previous research suggests that it is less likely to occur in the wild because haplotypes are uncommon between guanacos and vicuñas", but are there shared or private mitochondrial haplotypes between Lama glama and Lama guanicoe?

5) add more keywords, for example the name of the species, domestication, mitochondrial DNA, the geographic area, etc,

6) in the caption of the Figure 3 add an explanation about the use of the colors blue, red, green and black for the samples'ID.

Minor edits

The paper will benefit from a revision of the English, I suggested here some words or sentences that need to be revised.

- the term "presense" need to be changed in "presence" (e.g. lines 59, 67, 90, 92, etc)

- line 61 " Traditionally, In South..." change in "Traditionally, in South...

- check "reducancy" at line 245

- check the concentration of the library (9 ng/ml) line 211

- line 252 check the sentence "41 sampled ancient Holocene samples sequenced" and specify that 41 are referred to the samples analysed in the paper.

- check "socieites" at line 351

- line 389 check "intead"

6. PLOS authors have the option to publish the peer review history of their article (what does this mean?). If published, this will include your full peer review and any attached files.

Reviewer #1: No

Reviewer #2: No

---

## [Author Response · Author response to Decision Letter 0]

12 Jun 2020

REVIEWER: 1

I have to signal only a couple of typos: line 90 "presence" and S1 Table "Provenience"

The line 90 "presence" and S1 Table with "Provenience" was corrected the typos

I would like to see in S1 Table at least the median length of the sequenced molecules mapping on the target.

We have added in S2 Table (samples sequenced in this study, S1 Table belong to sequences used from GenBank) a column with number fragments mapped to reference, total length of mapped fragments and average fragment length 

I also suggest to spend some words to briefly describe the DNA extraction method, since it is not clear only through the cited reference.

The reviewer must have missed the information in the original submission. We describe the DNA extraction method in detail from lines 190 to 217. 

It is a pity that the authors could not analyse any sample from Site LD-S4 that they mentioned as the one with evidence of domestic camelids according to osteometric evidence. Were those samples not available for the analysis?

We agree with the reviewer; we could not obtain the permission. 

REVIEWER: 2

1) In the abstract it would be better to indicate the number of samples for which the authors obtained results (41?) and not the total samples analysed (52?), or at least to specify that.

It was corrected

2) In Table 1 I think that the authors should: 

(i) add a colum where specify what kind of skeletal element was analysed (e.g. petrous bone, long bone, tooth...); It was added

(ii) add a column where they indicate if the chronology is based on radiocarbon dating or material culture; It was described below Table 1 

(iii) check the number of samples, because in the abstract and results the total number of samples is 53 and the ones with sufficient endogenous is 41, but in the table 1 there are 53 samples, of which 10 with sufficient endogenous; 

(iv) explain why some samples have the same ID, are they referred to the same individual?

(iii and iv) 19, 78, 31 were done duplicated for NGS.

41 was used for phylogenetic analyses, using the sequence available for two loci, specifically cytb (1,140 bp) and D-loop (1,215 bp)

3) The authors followed proper ancient DNA guidelines, but I suggest to add the results from MapDamage software to analyzse damage patterns with the aim to authenticate the data here obtained.

While we agree with the reviewer that being able to visualize damage patterns along mapped reads using a program such as MapDamage is helpful, we are unable to employ due to the method used for mapping. We used the mapping iterative assembler (MIA, https://github.com/mpieva/mapping-iterative-assembler) for mapping merged reads to the mtDNA reference genome, which does not generate the necessary file type required for running MapDamage (SAM/BAM format). MIA was originally developed for working specifically with ancient DNA (i.e., Neanderthal mtDNA) and provides a number of additional steps that help minimize erroneous SNP calls when working with ancient DNA. For example, the method uses a position specific substitution matrix that has been altered to work with damaged/ancient DNA to help reduce potential calling errors. The method will iterate until convergence on a consensus sequence, which is one of the main selling points of this method when mapping ancient mtDNA.

To further reduce potential erroneous SNP calls associated with cytosine deamination, “total DNA was treated with 1X USER enzyme mix (New England BioLabs) for 3h at 37°C to reduce nucleotide substitution errors …., which is common when working with aDNA [74,75]” (line 211). This mix combines two enzymes, the Uracil-DNA glycosylase and DNA glycosylase-lyase endonuclease VIII and replaces with abasic sites the cytosines that have been deaminated post-mortem into uracils. So, the transitions C-T has been corrected, which is what MapDamage is used to show when not corrected. 

4) Given that the identification of domestic from wild species through DNA analysis is often complicated and confused because of recent times of divergence and/or hybridisation phenomena, the paper will benefit from some more sentences about the power of mitochondrial DNA analysis in the identification of Lama glama and Lama guanicoe. For example, the authors stated that "previous research suggests that it is less likely to occur in the wild because haplotypes are uncommon between guanacos and vicuñas", but are there shared or private mitochondrial haplotypes between Lama glama and Lama guanicoe?

There are sufficient differences between llama and guanaco at the cytb and D-loop to result in reciprocal monophyly in the gene trees as shown on Fig 3. In contrast to previous studies, we have used both markers, the cytb and D-loop, together increasing the conserved sites and maintaining the polymorphisms. 

 “As shown previously using mtDNA data [84], guanaco form a monophyletic clade separate from vicuña and domestic llama...” (line 293)

5) add more keywords, for example the name of the species, domestication, mitochondrial DNA, the geographic area, etc,

The suggestions have been done

6) in the caption of the Figure 3 add an explanation about the use of the colors blue, red, green and black for the samples'ID.

“Sample names are color coded according to whether the ancient sample was sequenced for this study (blue) and originated near the high altitude LD-S4 site (red), or were contemporary (black) or ancient (green) mtDNA sequences obtained from GenBank.” (Fig 3)

Minor edits

The paper will benefit from a revision of the English, I suggested here some words or sentences that need to be revised.

- the term "presense" need to be changed in "presence" (e.g. lines 59, 67, 90, 92, etc)

- line 61 " Traditionally, In South..." change in "Traditionally, in South...

- check "reducancy" at line 245 the word is “reduced”

- check the concentration of the library (9 ng/ml) line 211

- line 252 check the sentence "41 sampled ancient Holocene samples sequenced" and specify that 41 are referred to the samples analysed in the paper.

- check "socieites" at line 351

- line 389 check "intead"

All of the above indicated edits have been implemented in the revised manuscript.

These changes have clearly improved our manuscript. Thank you again for consideration of our revised manuscript.

---

## [Decision Letter · Decision Letter 1]

26 Aug 2020

PONE-D-20-06090R1

Were domestic camelids present on the prehispanic South American agricultural frontier? An ancient DNA study

PLOS ONE

Dear Dr. Abbona,

Thank you for submitting your manuscript to PLOS ONE. After careful consideration, we feel that it has merit but does not fully meet PLOS ONE’s publication criteria as it currently stands. Therefore, we invite you to submit a revised version of the manuscript that addresses the points raised during the review process.

Reviewer 1 asks you to correct a few minor issues before the publication. These are listed below. 

Reviewer 3 has some issues with your choice to map to the guanaco mtDNA reference genome for the phylogenetic reconstructions. Furthermore, the reviewer thinks that the discussion and conclusions sections should be edited to primarily emphasize the contributions of this study. Please, also check the reviewer's suggestions concerning the figures. There is also a list of minor issues that should be addressed. 

Please submit your revised manuscript by October 10th 2020 If you will need more time than this to complete your revisions, please reply to this message or contact the journal office at plosone@plos.org. Please include the following items when submitting your revised manuscript:

We look forward to receiving your revised manuscript.

Kind regards,

Mario Novak

Academic Editor

PLOS ONE

Reviewers' comments:

Reviewer's Responses to Questions

**Comments to the Author**

1. If the authors have adequately addressed your comments raised in a previous round of review and you feel that this manuscript is now acceptable for publication, you may indicate that here to bypass the “Comments to the Author” section, enter your conflict of interest statement in the “Confidential to Editor” section, and submit your "Accept" recommendation.

Reviewer #2: All comments have been addressed

Reviewer #3: (No Response)

2. Is the manuscript technically sound, and do the data support the conclusions?

Reviewer #2: (No Response)

Reviewer #3: Partly

3. Has the statistical analysis been performed appropriately and rigorously? 

Reviewer #2: (No Response)

Reviewer #3: Yes

4. Have the authors made all data underlying the findings in their manuscript fully available?

Reviewer #2: (No Response)

Reviewer #3: Yes

5. Is the manuscript presented in an intelligible fashion and written in standard English?

Reviewer #2: (No Response)

Reviewer #3: Yes

6. Review Comments to the Author

Reviewer #2: The authors addressed all the issues raised, however there are some minor issues to be checked and fixed before the publication:

- l.46 the term "reamain"

- Table 1 the terms: "Izqenon", "Cuneiforme" and "Cuboide"

- l. 201 "Dremmel"

l. 202 "negative" hairflow hood. The fact that it has negative pressure is in contrast with the criteria of aDNA analysis.

Reviewer #3: I really enjoyed reading this manuscript. While I was previously unfamiliar with the specific question this study addresses—whether domestic camelids were present on the pre-Hispanic South American agricultural frontier—the authors provided ample context to explain the importance of this question to non-experts. Further they pose a question—whether camelid remains in their study area belong to domesticate or camelid species—that is well suited to ancient mitochondrial DNA analysis, and they provide strong support for why other methods (such as morphological assessment) cannot sufficiently address this question. In their analyses, the authors present compelling evidence that weakens previous claims that domestic camelids may have been present in their study area during the pre-Hispanic period. Overall, I think that this is a nice study that will be of interest to archaeologists and to those curious about camelid domestication in the Americas generally.

I do have several suggestions that I think would help strengthen the paper, which are listed below.

Broad comments-

Reference genome choice –

I am concerned by the choice to map to the guanaco mtDNA reference genome for the phylogenetic reconstructions. While the authors state that this choice was made because the maximum coverage was achieved when this reference genome was used (lines 290-296), I worry that this will introduce bias into the analysis, the aim of which is to distinguish between guanaco, llama and vicuña. Ideally, I think it would be best to map to a reference genome from an outgroup species (I will leave it to the authors to determine what the best outgroup species might be, perhaps the Arabian or Bactrian camel). If it isn’t possible to replicate this analysis using a reference genome from a true outgroup species, I think it would be valuable for the authors to demonstrate whether they see consistent results when the llama and vicuña mitochondrial reference genomes are used.

Structure –

While the authors do an excellent job placing their study within the context of existing literature, I think that the discussion and conclusions sections should be edited to primarily emphasize the contributions of this study. Of the 12 paragraphs included in the discussion, only 4 directly discuss findings from this study. Similarly, only about half of the conclusion section is directly focused on explaining this study’s significance. This may make it difficult for readers to understand what the specific contributions of this study are to the field. I would therefore recommend moving many of these background details to the introduction and reworking the discussion and conclusion sections to focus more specifically on interpreting the findings of this study. That is not to say that these findings should not be placed within the context of findings from previous studies, but I think it would be best if these studies were already introduced in the introduction so that references to them in the discussion section can be more focused.

Figures-

Figure 1a – This map conveys really important information, but I find it quite difficult to read. I think it would be best to remake this figure using an underlying map that is less detailed, so that the things the authors are trying to highlight stand out more clearly. I think it would be sufficient to outline the study area with an unfilled box, as the patterned shading used makes it difficult to see the text within it. Also, I think that the dashed lines and labels at the top of the figure showing the limits of southern domestication are easy to miss, because the dashed lines blend in with the map and the white text is difficult to see. I also think it could be useful for the entire map to be zoomed out slightly, so that these limits and the southern-most point in the study area are not all so close to the edges of the map. I do really appreciate the insert showing the location of this map within the larger map of South America, so please keep something like this.

Here are some suggestions of free mapping tools that include simple maps that you might find useful. I only mention them in case you don’t already have a method of map making that you are happy with: Tableau (Academic version), ggmaps (in R), basemaps (in python), Simplemappr.

Figure 1b – Would it be possible for the authors to recreate this map? While it conveys interesting information, it’s quite blurry and the image quality feels a bit out of place.

Figure 2 – I think that this map should be combined with figure 1a, with all the points highlighted in it just inserted into the same map in figure 1a. I think that if a less detailed base map is used, the sites highlighted in this plot will also be easier to see. Right now, the sites that fall on top of high elevation areas are difficult to see.

Figure 3 – The use of red and green colors to distinguish between samples is not accessible to readers with red/green color blindness. Additionally, all colors will be lost to any reader who prints out the study on a black and white printer. Perhaps you could add some sort of differentiating symbols next to the colored names to help distinguish them from one another.

Line by line comments-

Line 54 – masl is a non-standard abbreviation. I assume it stands for meters above sea level, but please write this out in full the first time you use it and then explicitly define the acronym (or since you don’t use the term that often, I think you could just write it out fully each time).

Line 61 – “and the isotopic signal of δ13C and δ15N from human remains” – this phrasing is a little odd. I think it sounds like humans don’t usually have any sort of 13C or 15N signals. I suggest “and δ13C and δ15N signatures measured in human remains”

Line 63 – RCYBP is a non-standard acronym, so please write it out in full the first time you use it (I don’t think it is referenced again, so possibly there is no need to include an acronym).

Line 122 – I think it would be beneficial to add a line explicitly stating that no remains from LD-S4 were analyzed in this study. Since the site is referenced heavily, I didn’t initially realize that it wasn’t the focus of this study.

Line 126 – “25% (n=3 samples) of the total faunal assemblage (n=12 samples)” I think this line can be simplified to read: “25% of the total faunal assemblage (n=12 samples)”

Lines 158-178 – I think it would be helpful to more explicitly state that you are analyzing ancient mitochondrial DNA and discussing its specific value in this section. With the increasing prevalence of genome-wide ancient DNA studies, I think that readers don’t know what kind of analysis to expect when just the term ancient DNA is used.

Line 173 – “ancient DNA” should be abbreviated to “aDNA” since this abbreviation has already been established

Line 180 – Should read “Materials and Methods”

Line 181-184- “Fifty-two camelid bone samples identified to Lama spp. based on osteological characters were processed from 25 cultural assemblages from 17 archaeological sites located in central and southern Mendoza (Fig 2), which represents 70% of the available archaeological sites with faunal remains from the region.” This phrasing is a little awkward, I suggest “Fifty-two camelid bone samples (identified as Lama spp. based on osteological characters) from 25 cultural assemblages excavated from 17 archaeological sites located in central and southern Mendoza were selected for analysis. This represents 70% of the available archaeological sites with faunal remains from the region”

Table 1 – I recommend changing the column heading “Functionality” to “Site function.” The following words are not in English: Izqueon (Ischium), Cuneiforme (Cuneiform), Cuboide (Cuboid). Line 192- I am not sure what the authors mean when they state that samples 19, 78 and 31 were duplicated, especially since I count 50 rows in the table. Please clarify this.

Line 204 – Please explicitly state what methods were used to specifically minimize contamination with contemporary DNA. There is a lot of variation between labs, so this is helpful to know.

Line 206-207 – The paper cited here is actually a comparison of multiple methods, so this is confusing. Also, since the next lines are about sampling methods and not DNA extraction, I think it may be appropriate to make a more general statement about the ancient DNA processing methods used, or to remove this sentence.

Line 209-210 – Please specify how long bones were UV crosslinked for.

Line 218 – I don’t think ref 74 is the correct citation. This paper lists multiple extraction methods, all of which are described in other papers, and I don’t think they involve a pre-treatment. Please list the original paper that this pretreatment was based on and list any modifications if necessary.

Line 227 – 229 – I think this sentence should be moved to the next paragraph, as UDG treatment is typically considered part of the library preparation. Also, while the citations provided do discuss ancient DNA damage, I don’t think that either are the original description of the UDG treatment protocol used, please add this citation.

Line 231-232 – What method was used to ensure that this starting concentration was used? If you just used a standard volume of extract, please specify that instead.

Line 242 – Please write out this abbreviation in full (I assume it stands for Paired End)

Line 244 – I don’t think it is necessary to provide the submission ID

Line 248-249 – Please state what criteria were used to merge paired end overlapping reads (i.e. how much overlap was required, was there any allowance for mismatch within this overlapping region made)

Line 253-254 – “The resulting mitochondrial genomes were assessed by visual inspection” – please specify what this process entailed

Line 261-262- Please provide a reference for this statement if possible.

Line 264-267 – Ancient DNA C-to-T damage is another source of error that could impact this (note that while UDG treatment was used to remove this damage, it cannot fix damage that occurs at CpG sites).

Line 275 – Change “previous” to “previously”

Line 277-279 – Please specify all default parameters were chose or if any modifications were made in this process

Line 281-283 - Please specify all default parameters were chose or if any modifications were made in this process

Line 290-291 – Please state what criteria were used to define successful recovery

Line 353 – change “chrososome” to “chromosomes”

Line 354 – I’m not sure if C and G banding patterns are very commonly discussed anymore. I would recommend either leaving this reference out or describe what they are and their significance for readers who might be unfamiliar with this term.

Line 354-355 – The phrasing “DNA nucleotide methods” is a bit odd. I recommend saying “Genetic methods”

Line 370 – I think this should this read “because shared haplotypes are uncommon” otherwise I am not sure what the meaning of this sentence is.

Line 373 – “Crossbreeding among those species, however, produces fertile hybrids in captivity [70,95]. Although hybridization between domestic species is possible, it does not interfere with sample identification.” I’m not sure if these lines are necessary

Line 408-413—“Osteometric evidence supporting the presence of domestic camelids is from bone specimens from a single high elevation village from southern Mendoza (Site LD-S4; [15]. Six DNA samples from camelids obtained from three others high elevation sites (El Indígeno, Los Peuquenes and Risco de los Indios) that are in close geographic proximity to LD-S4 and that date to approximately the same period [26,97] were included in the analysis.” I am not sure that this statement is clear enough that no samples from LD-S4 were included in this study. I recommend rephrasing this paragraph to make this clearer. For example: “Previous osteometric analyses of bone specimens from a single high elevation village from southern Mendoza (Site LD-S4; [15]) support the presence of domestic camelids in this region. While we were unable to include these samples in our study, six DNA samples from camelids obtained from three other high elevation sites (El Indígeno, Los Peuquenes and Risco de los Indios) that are in close geographic proximity to LD-S4 and that date to approximately the same period [26,97] were included in the analysis.”

Line 424 -429 – I am curious what the alternative conclusions for how these remains arrived at these archaeological sites if they were not from domesticated llama. I assume that the alternative hypothesis is that local people hunted wild guanaco, but it might be nice to specifically state this (assuming that it is true).

Table 2- Please provide more information about the meaning of the abbreviations masl, SL and WL. Additionally, I’m confused by the blank columns labeled “guanaco alignment” and “llama alignment.” Are these headers for the columns to the right? If so, could these titles instead be placed above the corresponding columns.

7. PLOS authors have the option to publish the peer review history of their article (what does this mean?). If published, this will include your full peer review and any attached files.

Reviewer #2: No

Reviewer #3: **Yes: **Éadaoin Harney

---

## [Author Response · Author response to Decision Letter 1]

9 Sep 2020

REVIEWER: 2

The authors addressed all the issues raised, however there are some minor issues to be checked and fixed before the publication

l.46 the term "reamain"

Table 1 the terms: "Izqenon", "Cuneiforme" and "Cuboide"

l. 201 "Dremmel" 

l. 202 "negative" hairflow hood. 

We thank the reviewer for identifying the above errors. All of the above indicated edits have been implemented in the revised manuscript.

REVIEWER: 3

Reference genome choice –

I am concerned by the choice to map to the guanaco mtDNA reference genome for the phylogenetic reconstructions. While the authors state that this choice was made because the maximum coverage was achieved when this reference genome was used (lines 290-296), I worry that this will introduce bias into the analysis, the aim of which is to distinguish between guanaco, llama and vicuña. Ideally, I think it would be best to map to a reference genome from an outgroup species (I will leave it to the authors to determine what the best outgroup species might be, perhaps the Arabian or Bactrian camel). If it isn’t possible to replicate this analysis using a reference genome from a true outgroup species, I think it would be valuable for the authors to demonstrate whether they see consistent results when the llama and vicuña mitochondrial reference genomes are used.

We understand the reviewer’s concern with our use of the guanaco mtDNA reference genome for mapping since that could introduce ascertainment bias. However, the consensus sequences generated for each sample using the guanaco, llama, and vicuña mtDNA reference genomes separately were identical. We reported that we used the consensus sequences mapped to the guanaco mtDNA genome due to their higher coverage compared to the sequences generated from the other two reference genomes. We think that the above is sufficient to address concerns about possible ascertainment bias, and was one of the options suggested by the reviewer. (line 364)

Structure –

While the authors do an excellent job placing their study within the context of existing literature, I think that the discussion and conclusions sections should be edited to primarily emphasize the contributions of this study. Of the 12 paragraphs included in the discussion, only 4 directly discuss findings from this study. Similarly, only about half of the conclusion section is directly focused on explaining this study’s significance. This may make it difficult for readers to understand what the specific contributions of this study are to the field. I would therefore recommend moving many of these background details to the introduction and reworking the discussion and conclusion sections to focus more specifically on interpreting the findings of this study. That is not to say that these findings should not be placed within the context of findings from previous studies, but I think it would be best if these studies were already introduced in the introduction so that references to them in the discussion section can be more focused.

We did several things to respond to this comment; however, we did not move portions of the discussion to an earlier part of the paper. The introduction efficiently provides the reader with the context of the study and flows well into the methods. However, we do appreciate that the reader did not like the structure of the discussion and conclusion. Thus, 1) we moved the two final paragraphs of the discussion to the beginning of that section as they state the direct implications of the study (we edited those paragraphs to fit there); 2) we edited the other paragraphs and added better topic and transition sentences to refer more directly to the results section; thus, the regional implications are clearly linked to the broader implications of the study. This helps the reader follow why each part of the discussion matters in the paper. 3) We deleted the first paragraph of the conclusion section because it simply reiterated in summary form what was just stated in the discussion. This will reduce that form of distraction for the reader. The conclusion now reiterates the central claims of the paper. We feel the discussion and conclusion sections are dramatically improved from these revisions. 

Figures-

Figure 1a – This map conveys really important information, but I find it quite difficult to read. I think it would be best to remake this figure using an underlying map that is less detailed, so that the things the authors are trying to highlight stand out more clearly. I think it would be sufficient to outline the study area with an unfilled box, as the patterned shading used makes it difficult to see the text within it. Also, I think that the dashed lines and labels at the top of the figure showing the limits of southern domestication are easy to miss, because the dashed lines blend in with the map and the white text is difficult to see. I also think it could be useful for the entire map to be zoomed out slightly, so that these limits and the southern-most point in the study area are not all so close to the edges of the map. I do really appreciate the insert showing the location of this map within the larger map of South America, so please keep something like this. 

Here are some suggestions of free mapping tools that include simple maps that you might find useful. I only mention them in case you don’t already have a method of map making that you are happy with: Tableau (Academic version), ggmaps (in R), basemaps (in python), Simplemappr.

We thank the reviewer for the suggested edits. We have modified the map to make it easier to read as requested by the reviewer.

Figure 1b – Would it be possible for the authors to recreate this map? While it conveys interesting information, it’s quite blurry and the image quality feels a bit out of place.

We have recreated this map to make it easier to read.

Figure 2 – I think that this map should be combined with figure 1a, with all the points highlighted in it just inserted into the same map in figure 1a. I think that if a less detailed base map is used, the sites highlighted in this plot will also be easier to see. Right now, the sites that fall on top of high elevation areas are difficult to see.

Figure 2 was combined with Figure 1A.

Figure 3 – The use of red and green colors to distinguish between samples is not accessible to readers with red/green color blindness. Additionally, all colors will be lost to any reader who prints out the study on a black and white printer. Perhaps you could add some sort of differentiating symbols next to the colored names to help distinguish them from one another.

We thank the reviewer for this request. The figure has been modified to address potential concerns with accessibility due to color choice.

Line by line comments – 

Line 54 – masl is a non-standard abbreviation. I assume it stands for meters above sea level, but please write this out in full the first time you use it and then explicitly define the acronym (or since you don’t use the term that often, I think you could just write it out fully each time).

We thank the reviewer for catching this error. We have now written the abbreviation in full the first time it is used in the manuscript. (line 53 and 443)

Line 61 – “and the isotopic signal of δ13C and δ15N from human remains” – this phrasing is a little odd. I think it sounds like humans don’t usually have any sort of 13C or 15N signals. I suggest “and δ13C and δ15N signatures measured in human remains”

The suggested edit was used in the revised manuscript (line 61)

Line 63 – RCYBP is a non-standard acronym, so please write it out in full the first time you use it (I don’t think it is referenced again, so possibly there is no need to include an acronym).

Thank you again! We have made the suggested edit by specifying “radiocarbon years before present” (line 63)

Line 122 – I think it would be beneficial to add a line explicitly stating that no remains from LD-S4 were analyzed in this study. Since the site is referenced heavily, I didn’t initially realize that it wasn’t the focus of this study.

We have added a sentence at the end of the Introduction stating “Remains from LD-S4 were not analyzed in this study because we could not obtain the permission” (line 204)

Line 126 – “25% (n=3 samples) of the total faunal assemblage (n=12 samples)” I think this line can be simplified to read: “25% of the total faunal assemblage (n=12 samples)”

We have used the reviewer’s suggested edit in the revised manuscript (line 136)

Lines 158-178 – I think it would be helpful to more explicitly state that you are analyzing ancient mitochondrial DNA and discussing its specific value in this section. With the increasing prevalence of genome-wide ancient DNA studies, I think that readers don’t know what kind of analysis to expect when just the term ancient DNA is used.

We have added ancient mitochondrial DNA (Line 187)

Line 173 – “ancient DNA” should be abbreviated to “aDNA” since this abbreviation has already been established

We have made the suggested change here and throughout the revised manuscript (line 203)

Line 180 – Should read “Materials and Methods”

We have made the suggested change (line 211)

Line 181-184- “Fifty-two camelid bone samples identified to Lama spp. based on osteological characters were processed from 25 cultural assemblages from 17 archaeological sites located in central and southern Mendoza (Fig 2), which represents 70% of the available archaeological sites with faunal remains from the region.” This phrasing is a little awkward, I suggest “Fifty-two camelid bone samples (identified as Lama spp. based on osteological characters) from 25 cultural assemblages excavated from 17 archaeological sites located in central and southern Mendoza were selected for analysis. This represents 70% of the available archaeological sites with faunal remains from the region”

We have used the reviewer’s suggested edit in the revised manuscript (lines 212-215)

Table 1 – I recommend changing the column heading “Functionality” to “Site function.” The following words are not in English: Izqueon (Ischium), Cuneiforme (Cuneiform), Cuboide (Cuboid). Line 192- I am not sure what the authors mean when they state that samples 19, 78 and 31 were duplicated, especially since I count 50 rows in the table. Please clarify this.

We have made the suggested changes and have deleted the statement concerning duplicated samples as that was an error. 

Line 204 – Please explicitly state what methods were used to specifically minimize contamination with contemporary DNA. There is a lot of variation between labs, so this is helpful to know.

We described in the methods used to specifically minimize contamination with contemporary DNA. For example, each bone sample was ground into a powder using a freezer mill with a sterile grinding vial. The aDNA laboratory is dedicated solely for work on extracting DNA from ancient samples where no camelid samples had been processed prior to this study. Extractions was conducted in groups of fifteen that included a negative control in each group to verify no cross-contamination or reagent contamination. Strict protocols for working with aDNA were employed to eliminate possibilities for contamination with other DNAs and all equipment was sterilized between use with samples using multiple methods (e.g. 10% bleach, 95% ethanol, and UV crosslinker). The NGS library preparation was carry out in a separate lab which also had not been exposed to any camelid samples prior to this study.

Line 206-207 – The paper cited here is actually a comparison of multiple methods, so this is confusing. Also, since the next lines are about sampling methods and not DNA extraction, I think it may be appropriate to make a more general statement about the ancient DNA processing methods used, or to remove this sentence.

The reviewer is correct that the reference paper compares three different methods, and we have modified the manuscript by referencing the original paper describing the extraction method used in this study (Yang et al. 1998) with specific modifications as described in our methods section. Some of the modifications we also used in Gamba et al. 2014 and 2016, which is why we originally cited the latter reference since it is the most similar in terms of how we conducted our extractions. We agree with the reviewer, however, that it is appropriate to cite the original paper describing the method, which we have now done (line 256-257). 

Line 209-210 – Please specify how long bones were UV crosslinked for.

We have added the requested information to the revised manuscript (line 260)

Line 218 – I don’t think ref 74 is the correct citation. This paper lists multiple extraction methods, all of which are described in other papers, and I don’t think they involve a pre-treatment. Please list the original paper that this pretreatment was based on and list any modifications if necessary.

The reviewer is correct. We have no referenced the original paper (Yang et al. 1998) and described the modifications where appropriate.

Line 227 – 229 – I think this sentence should be moved to the next paragraph, as UDG treatment is typically considered part of the library preparation. Also, while the citations provided do discuss ancient DNA damage, I don’t think that either are the original description of the UDG treatment protocol used, please add this citation.

The indicated citation was added (line 285)

Line 231-232 – What method was used to ensure that this starting concentration was used? If you just used a standard volume of extract, please specify that instead.

We apologize that the specified information was not included in the previous version of the manuscript. DNA concentrations for all extracted samples was measured with a Qubit Fluorometer using a dsDNA HS Assay Kit (line 285). 

Line 242 – Please write out this abbreviation in full (I assume it stands for Paired End)

The reviewer was correct, and we have now added the requested information to the revised manuscript (line 298)

Line 244 – I don’t think it is necessary to provide the submission ID

We deleted the submission ID as requested (line 299)

Line 248-249 – Please state what criteria were used to merge paired end overlapping reads (i.e. how much overlap was required, was there any allowance for mismatch within this overlapping region made)

This section of the methods has been modified. Default settings in SeqPrep were used to merge PE overlapping reads (line 303)

Line 253-254 – “The resulting mitochondrial genomes were assessed by visual inspection” – please specify what this process entailed

We agree with the reviewer that this statement was ambiguous. We have now deleted this sentence from the revised manuscript.

Line 261-262- Please provide a reference for this statement if possible.

The indicated sentence was not included in the revised manuscript.

Line 275 – Change “previous” to “previously”

We have made the suggested change (line 331)

Line 277-279 – Please specify all default parameters were chose or if any modifications were made in this process and Line 281-283 - Please specify all default parameters were chose or if any modifications were made in this process

We have included additional information as requested (line 341-352)

Line 290-291 – Please state what criteria were used to define successful recovery

“Successfully” was deleted (line 349)

Line 353 – change “chrososome” to “chromosomes”

We have corrected the misspelling as identified (line 458)

Line 354 – I’m not sure if C and G banding patterns are very commonly discussed anymore. I would recommend either leaving this reference out or describe what they are and their significance for readers who might be unfamiliar with this term.

C and G banding patterns method is cheaper than NGS, so if it was possible to differentiate them through banding patterns it was convenient.

Line 354-355 – The phrasing “DNA nucleotide methods” is a bit odd. I recommend saying “Genetic methods”

We changed for Sequencing methods (because Cytogenetic is also a genetic method)

Line 370 – I think this should this read “because shared haplotypes are uncommon” otherwise I am not sure what the meaning of this sentence is.

We have made the suggested change as indicated (line 483)

Line 373 – “Crossbreeding among those species, however, produces fertile hybrids in captivity [70,95]. Although hybridization between domestic species is possible, it does not interfere with sample identification.” I’m not sure if these lines are necessary

The indicated sentences were not included in the revised manuscript.

Line 408-413—“Osteometric evidence supporting the presence of domestic camelids is from bone specimens from a single high elevation village from southern Mendoza (Site LD-S4; [15]. Six DNA samples from camelids obtained from three others high elevation sites (El Indígeno, Los Peuquenes and Risco de los Indios) that are in close geographic proximity to LD-S4 and that date to approximately the same period [26,97] were included in the analysis.” I am not sure that this statement is clear enough that no samples from LD-S4 were included in this study. I recommend rephrasing this paragraph to make this clearer. For example: “Previous osteometric analyses of bone specimens from a single high elevation village from southern Mendoza (Site LD-S4; [15]) support the presence of domestic camelids in this region. While we were unable to include these samples in our study, six DNA samples from camelids obtained from three other high elevation sites (El Indígeno, Los Peuquenes and Risco de los Indios) that are in close geographic proximity to LD-S4 and that date to approximately the same period [26,97] were included in the analysis.”

We have changed the sentence as recommended by the reviewer (line 538-543)

Line 424 -429 – I am curious what the alternative conclusions for how these remains arrived at these archaeological sites if they were not from domesticated llama. I assume that the alternative hypothesis is that local people hunted wild guanaco, but it might be nice to specifically state this (assuming that it is true).

We agree with the reviewer that the stated alternative conclusion is possible, and have included a statement suggesting that possibility in the revised manuscript (line 568)

Table 2- Please provide more information about the meaning of the abbreviations masl, SL and WL. Additionally, I’m confused by the blank columns labeled “guanaco alignment” and “llama alignment.” Are these headers for the columns to the right? If so, could these titles instead be placed above the corresponding columns.

We have now written the abbreviation in full (Table 2).

The authors all agree that the reviewers’ comments and subsequent changes have improved the quality of our manuscript. Thank you again for considering our revised manuscript for publication. 

Best regards,

---

## [Decision Letter · Decision Letter 2]

28 Sep 2020

Were domestic camelids present on the prehispanic South American agricultural frontier? An ancient DNA study

PONE-D-20-06090R2

Dear Dr. Abbona,

We’re pleased to inform you that your manuscript has been judged scientifically suitable for publication and will be formally accepted for publication once it meets all outstanding technical requirements.

Kind regards,

Mario Novak

Academic Editor

PLOS ONE

Additional Editor Comments (optional):

Reviewers' comments:

Reviewer's Responses to Questions

**Comments to the Author**

1. If the authors have adequately addressed your comments raised in a previous round of review and you feel that this manuscript is now acceptable for publication, you may indicate that here to bypass the “Comments to the Author” section, enter your conflict of interest statement in the “Confidential to Editor” section, and submit your "Accept" recommendation.

Reviewer #3: All comments have been addressed

2. Is the manuscript technically sound, and do the data support the conclusions?

Reviewer #3: (No Response)

3. Has the statistical analysis been performed appropriately and rigorously? 

Reviewer #3: (No Response)

4. Have the authors made all data underlying the findings in their manuscript fully available?

Reviewer #3: (No Response)

5. Is the manuscript presented in an intelligible fashion and written in standard English?

Reviewer #3: (No Response)

6. Review Comments to the Author

Reviewer #3: (No Response)

7. PLOS authors have the option to publish the peer review history of their article (what does this mean?). If published, this will include your full peer review and any attached files.

Reviewer #3: **Yes: **Eadaoin Harney

---

## [Editor Report · Acceptance letter]

20 Oct 2020

PONE-D-20-06090R2 

Were domestic camelids present on the prehispanic South American agricultural frontier? An ancient DNA study 

Dear Dr. Abbona:

I'm pleased to inform you that your manuscript has been deemed suitable for publication in PLOS ONE. Congratulations! Your manuscript is now with our production department. 

Kind regards, 

on behalf of

Dr. Mario Novak 

Academic Editor

PLOS ONE